# Effects of Printing Parameters on the Fatigue Behaviour of 3D-Printed ABS under Dynamic Thermo-Mechanical Loads

**DOI:** 10.3390/polym13142362

**Published:** 2021-07-19

**Authors:** Feiyang He, Muhammad Khan

**Affiliations:** 1School of Aerospace, Transport and Manufacturing, Cranfield University, College Road, Cranfield MK43 0AL, UK; 2Centre for Life-Cycle Engineering and Management, Cranfield University, College Road, Cranfield MK43 0AL, UK; Muhammad.A.Khan@cranfield.ac.uk

**Keywords:** 3D printing, ABS, fatigue, thermo-mechanical loads, building orientation, nozzle size, layer thickness

## Abstract

Fused deposition modelling (FDM) is the most widely used additive manufacturing process in customised and low-volume production industries due to its safe, fast, effective operation, freedom of customisation, and cost-effectiveness. Many different thermoplastic polymer materials are used in FDM. Acrylonitrile butadiene styrene (ABS) is one of the most commonly used plastics owing to its low cost, high strength and temperature resistance. The fabricated FDM ABS parts commonly work under thermo-mechanical loads in actual practice. For producing FDM ABS components that show high fatigue performance, the 3D printing parameters must be effectively optimized. Hence, this study evaluated the bending fatigue performance for FDM ABS beams under different thermo-mechanical loading conditions with varying printing parameters, including building orientations, nozzle size, and layer thickness. The combination of three building orientations (0°, ±45°, and 90°), three nozzle sizes (0.4, 0.6, and 0.8 mm) and three-layer thicknesses (0.05, 0.1, and 0.15 mm) were tested at different environmental temperatures ranging from 50 to 70 °C. The study attempted to find the optimal combination of the printing parameters to achieve the best fatigue behaviour of the FDM ABS specimen. The experiential results showed that the specimen with 0° building orientation, 0.8 mm filament width, and 0.15 mm layer thickness vibrated for the longest time before the fracture at each temperature. Both a larger nozzle size and thicker layer height can increase the fatigue life. It was concluded that printing defects significantly decreased the fatigue life of the 3D-printed ABS beam.

## 1. Introduction

Three-dimensional (3D) printing, also known as additive manufacturing (AM), is a rapid prototyping technology that was developed in the 1980s. This technology is based on the computer-aided design (CAD) model and helps to construct physical objects using powdered metal or plastic and other bondable materials [1]. Fused deposition modelling (FDM), as an AM process, was initially developed in the early 1990s. It develops the components using layer by layer deposition with the thermoplastic material filaments extruded through a nozzle [2]. Due to its safe, fast, and effective operation, freedom of customisation, and cost-effectiveness, FDM has become the most widely used AM process in customised and low-volume production industries [3]. Apart from functional prototyping, FDM is also used in aerospace, automotive, medical and biomechanical sectors [4,5,6,7,8]. 

Several different thermoplastic polymer materials are used in FDM. Acrylonitrile butadiene styrene (ABS) is the most commonly used plastic because of its low expense, high strength, and temperature resistance [9]. Although the differences in the mechanical properties between FDM ABS and conventionally manufactured ABS have been discussed in the literature [10,11,12,13,14,15], the most common conclusion was that FDM ABS has 11% to 37% reduced modulus and 22% to 57% reduced strength [16,17]. Some studies evaluated the different printing parameter effects on FDM ABS mechanical properties (strength and modulus) [18,19,20,21,22,23]. Some studies attempted to improve the mechanical properties of FDM polymers by optimising the 3D printing parameters [13,24,25,26]. However, only a few studies focused on the fatigue behaviour of FDM ABS [10,27,28,29,30,31].

As a result of the propagation of cracks due to a repetitive or cyclic load, fatigue failure is normal in practical structures where the cyclic stresses are typically significantly less than yielding strength. Therefore, it is significant to evaluate the fatigue performance of the material. Safai et al. Shanmugam et al. and He et al. reviewed the fatigue behaviour of FDM polymeric materials [8,32,33]. The results of their overview show that determining the best printing parameter combinations for fatigue strength is a challenge because of the synergism between variables and 3D printed material properties. The experimental studies are essential to understand how these parameters affect the fatigue behaviour. Therefore, some studies carried out the experimental works for different 3D printed polymers with variable parameters. Letcher and Waytashek and Afrose et al. all reported that 45° building orientation generated the maximum fatigue strength for 3D printed polylactic acid (PLA) in tension fatigue tests [34,35,36]. The results of a tension fatigue test for FDM ABS had the same conclusion [10,28,29]. However, the 0°(X)/90°(Y) raster orientation had the best fatigue life in compact tension test for FDM PLA [37]. X orientation provided the highest fatigue strength for FDM ABS in a tension fatigue test [38]. For other printing parameters, some studies carried out the flexural fatigue tests for FDM PLA bar and the results showed that larger layer thickness and nozzle size both increased the fatigue life [39,40,41]. However, similar research has not been carried out on FDM ABS. Jap et al. listed the potential printing parameters, depicted in Figure 1 (dominant factors shown in red), which may affect the fatigue life of FDM ABS through many complex mechanisms and proposed that raster orientation is a critical parameter that is related to the tensile strength of FDM ABS [27]. Sharma and Ziemian proposed that the 0° or +45°/−45° raster orientation achieved the most fatigue cycles to failure in a tension–tension fatigue test [10,28,29]. By contrast, Hart and Wetzel confirmed that 90° laminae orientation had the highest fracture toughness [30]. Padzi et al. carried out the fatigue test for ABS dog bone specimens and concluded that the 3D printed part had a lower fatigue life [31].

After careful judgement, surprisingly, although there are several printing parameters that may affect the fatigue strength, only a few studies have tested the relationship between them and fatigue performance [10,27,28,29,30,31]; furthermore, these studies only investigated one or two specific parameters [38]. Safai et al. and Shanmugam et al. presented an overview of the fatigue behavior of 3D printed polymers. However, previous studies only performed the tension–tension or rotating bending fatigue tests for FDM ABS [32,33], and as such, although Puigoriol-Forcada et al. investigated the flexural fatigue behaviors of FDM polycarbon (PC) and proposed that build orientation significantly affected the fatigue life due to the inner anisotropy [42], there have been no studies on the bending/flexure fatigue test for FDM ABS. Additionally, no study has considered the thermo-mechanical loading conditions. 

The coupled thermo-mechanical load is more common in an actual working environment, such as the 3D printed flex sensor and capacitive interface device working under temperature load [43]. It even forms part of the development of the next-generation space exploration vehicle. NASA launched a FDM CubeSat Trailblazer in November 2013 to demonstrate the durability of 3D-printed devices with extreme thermal cycling [44]. We evaluated the relationship between the several different critical printing parameters and the bending fatigue performance for the FDM ABS beam under different thermo-mechanical loading conditions. Compared with the prior research, the extra thermo loads and dynamic bending stress simulated the practical working conditions better. Building orientation, nozzle size and layer thickness were selected as the significant variables in the research, because the adjustment of these parameters can affect the printing quality, which related to the air-void defects in the specimen. In the study, the specimens with different printing parameters were tested under bending stress with the thermo loads. The fatigue life was measured to find the optimal printing parameter combination to guide the future FDM process and provide a better fatigue life for the FDM structure.

## 2. Methodology

### 2.1. Design of Experiment

The current study focused on three significant parameters (building orientation, nozzle size and layer thickness) and investigated their critical effects on the mechanical behaviour of 3D-printed ABS, as already mentioned in the introduction. These parameters also affect the other aspects of the FDM process, such as the manufacturing time and cost.

Building orientation: this can be defined as the path on which the nozzle moves on each layer of the FDM part. It can be set in the 3D printer’s software;Nozzle size: this determines the diameter of the filament extruded from the printer’s nozzle. Different diameter nozzles can be replaced to print different specimens;Layer thickness: this defines the height of each layer and the number of layers each part has. It can be set in the 3D printer’s software.

Three values, which are typically used as the default setting for 3D printing, were evaluated for each parameter to cover the typical range of printing parameters. This facilitated a comprehensive evaluation of the experimental results. In addition, the nozzle size and layer thickness were derived from the profile of the default 3D. These values not only provided excellent printing quality but also covered the typical setting range. The selected values for each parameter are listed in Table 1 and depicted in Figure 2, Figure 3 and Figure 4.

Because the mechanical properties of ABS change a lot when temperature changes, while the practical applications may work under different environmental temperature, a comparison of fatigue life under different temperature is significant in the research. Therefore, three different environmental temperatures, 40, 50 and 60 °C, were selected for this purpose. 60 °C was chosen as the upper limit of temperature because ABS is sensitive to the temperature. The mechanical strength decreases when increasing temperature. The structural strength is not strong enough to support the experiment when environmental temperature is over 60 °C due to softening.

Because each printing parameter had three options as shown in Table 1, there were a total of 27 combinations of the printing parameters. Each combination was tested under three different temperatures, respectively. Consequently, 81 different types of specimen were printed for these parameters and the environmental temperature values. Three similar specimens were manufactured for each combination and tested under the same conditions for confirming the experimental repeatability of the results obtained.

### 2.2. Materials

In this study, a red ABS filament fabricated by Ultimaker^®^ (Utrecht, Netherlands) was chosen as the raw material because the crack growth path and plastic zones can be clearly observed in this colour. More details of this material are presented in Table 2.

### 2.3. Specimen Preparation

Currently, ASTM D7774 [48] is the standard test method for flexural fatigue properties of plastics. Following the standard, the specimen is cyclically loaded equally in the positive and negative directions to a specific stress or strain level at a uniform frequency until the specimen ruptures or yields. This means the quasi-static loads are applied in the test. 

However, the 3D-printed polymers, rather than conventional manufactured polymers, were tested in this research. Extra thermo loads and dynamic mechanical loads were applied during the tests. This means the current standard that focuses on the bending fatigue test of polymers may be not suitable in this research. Therefore, a 3D printing ABS specimen was designed using the CATIA v5 CAD software. The geometry of the specimen is depicted in Figure 5: 150 × 10 × 5 mm and was the same as that used in previous research [49]; therefore, the experimental results of the current study and those of the previous study can be compared. An initial seeded crack with a depth of 0.5 mm was crafted close to the fixed end of the beam to ensure that the maximum stress concentration occurred at the same locations for all the specimens. Therefore, crack propagation in all the experiments was the same and closely resembled fatigue failures of an actual scenario. The test results thus obtained were then compared.

The specimen’s CAD model was converted to an STL file and imported to the Ultimaker CURA 4.6 software. Apart from the selected parameters presented in Table 2, there were a series of parameters in CURA. Because these parameters in CURA were not the focus of the current study, all these settings were held at recommended or default values during the printing process. Further, an infill density of 100% was selected. The appropriate set nozzle temperature was selected as 245 °C based on the recommendation of the printing setup. The bed temperature was set to 90 °C, which was the default value of the printer. Finally, the Ultimaker 2+ printer was used to print all the specimens, as shown in Figure 6.

### 2.4. Experimental Scheme

The experiments were primarily divided into two parts. First, the number of cycles until the fracture of cantilever beams were evaluated using bending vibration tests. Then, dynamic mechanical analysis tests were performed for the specimens cut from the previous beams.

### 2.5. Experimental Setup and Procedure

The experimental setup is illustrated in Figure 7. The ABS specimen was fixed on the V55 shaker manufactured by Data Physics (Hailsham, UK). A mica band heater was used to provide the specimen’s constant thermal loads throughout the experiment. First, the specimen’s natural frequency was measured three times using an accelerometer in an impact test. The first-order natural frequency of the specimen can be calculated using Equation (1): (1)$$f=\frac{i-j}{t_{i}-t_{j}}$$
where $t_{i}$ and $t_{j}$ denote the time of the ith and jth peak amplitude, respectively. The vibration under natural frequency provided the initial maximum amplitude for the beam, and this significantly shortened the experiment time. The accelerometer recorded the acceleration and time data, imported them into Signal Express software via a DAQ card. The signal generator produced the sinusoidal output into the power amplifier. The power amplifier then transferred the signal into the shaker, following which the shaker vibrated with a 2 mm amplitude under the measured natural frequency of the beam. This implied that the beam was under resonation initially, following which the pre-seeded crack growth started. The shaker was running continuously until the beam’s final fracture occurred.

Compared with the traditional fatigue failure approach, which is defined as the number of cycles at which the stiffness of the material reduces by 50% (N_f50_), the current study directly registered the number of cycles from the beginning of the run to the occurrence of fracture for the specimens because it was difficult to measure the stiffness for the cracked specimen. The results were recorded and then compared with each other. The 3D printing parameter combination corresponding to most cycles was determined as the optimal choice. The beam section after the fracture was captured using a 200× Dino-Lite microscope (AnMo Electronics Corporation, Hsinchu, China).

The DMA test was then performed using Q800 from TA Instruments (New Castle, TX, USA). The specimen was fixed in the chamber of Q800 by the dual cantilever clamp. The temperature ramp rate was 3.00 °C/min. The load oscillation frequency was 1 Hz. This measured the storage modulus of the parts cut from the previous bending fatigue specimens under the temperature range of 30 to 70 °C.

## 3. Results and Discussion

### 3.1. DMA Tests

The storage modulus of each specimen cut from the cracked cantilever beams at temperature ranging from 30 to 70 °C was measured by DMA. The statistical significance was determined using MATLAB. A Kruskal–Wallis one-way analysis of variance (Kruskal–Wallis test) with three independent variables (building orientation, nozzle size and layer thickness) was conducted without the consideration of temperature because the data group did not meet the normal distribution requirement. Each printing parameter group contained the sample data, which was the constant in that parameter, but with the different values in other parameters. This tested the hypothesis that the storage modulus of different parameters were equal against the alternative hypothesis that at least one group was different from the others.

The variation of the storage modulus with temperature and statistical data is depicted in Figure 8 and Figure 9. The mean storage moduli for different printing parameters are listed in Table 3. Although calculating the mean storage modulus was not the primary purpose of this study, this value can be used as a reference since it significantly affects the performance of structural fatigue. Higher storage modulus means the higher stiffness. It may potentially lead to longer fatigue life.

As can be observed, the storage modulus always decreased with the increase in temperature during the DMA test for all the specimens. This is similar to other materials [49,50,51]. For the mean storage modulus under the different printing parameter settings, the X orientation provided the highest mean storage modulus, that is, 1692 MPa. The mean storage modulus for the specimen with the Y orientation was only 1561 MPa. The XY building orientation between these two orientations had a mean storage modulus of 1616 MPa. The variation in the storage modulus was found to be similar to that reported in previous studies [36,52,53,54].

The influence of the layer thickness appeared to be insignificant. Specimens with a layer thickness of 0.15 mm could provide a mean storage modulus of 1648 MPa, which is only approximately 60 MPa more than that provided by a 0.05 mm-thick layer specimen (1590 MPa). Whereas the results showed the difference when comparing with the previous works. Other previous research presented that lower layer thickness seemed to provide the higher strength [14,54]. However, the nozzle size was found to have a significant influence on the storage modulus. The specimens printed with 0.6 mm and 0.8 mm nozzles had a greatly improved storage modulus, that is, over 1700 MPa, as compared with the 1450 MPa mean storage modulus provided by the 0.4 mm nozzle size.

A statistical analysis of the data also indicated a similar conclusion. Compared with the *p*-value for different printing parameters, the *p*-value for the layer thickness of 2.30 × 10^−31^ was greater than the other two parameters. It confirmed that the storage modulus of FDM ABS is not significantly impacted with the change in layer thickness compared with other two parameters.

### 3.2. Bending Fatigue Test

Each group of three specimens was tested with a bending fatigue, and the mean cycles to fracture were recorded. The outliers inside each set were filtered and removed. Figure 10 shows the mean value of the number of cycles to the fracture counted for each parameter combination at different environmental temperatures. As can be seen, the specimen with an X raster orientation, nozzle size of 0.8 mm, and layer thickness of 0.15 mm printing had the longest fatigue life until the fracture at different environmental temperatures.

#### 3.2.1. Analysis of Variance

The fatigue life results were analysed through the Kruskal–Wallis test at a 95% confidence interval for each parameter. The study calculated the mean number of cycles for each constant parameter as one data group. The Kruskal–Wallis test tested the hypothesis that all group means are equal against the alternative hypothesis that at least one group is different from the others. It was used to determine whether the number of cycles from each group of parameters had a common mean. It can then find out whether different parameters have different influences on the beam’s fatigue life.

Table 4 lists the F and *p*-values for each parameter. The *p*-value for the building orientation is 5 × 10^−5^, for the nozzle size is 9.90 ×10^−4^, and for the layer thickness is 0.164, and for the temperature is 6.96 × 10^−11^.

Based on the *p*-values listed in the table above, we can confirm that temperature has the most significant influence on the fatigue strength, statistically. Building orientation and nozzle size also critically affect fatigue life. However, layer thickness did not show a significant influence on the range of values tested because the *p*-value 0.1684 exceeded 0.05.

#### 3.2.2. Temperature Influence on Fatigue Life

Figure 11 show the variation in the fatigue life at different environmental temperatures. As can be seen, temperature has a significant impact on fatigue life. Notably, all the curves show similar decreasing trends, confirming that the fatigue resistance decreases when the temperature increases from 50 to 70 °C. The prior research focusing on conventional manufactured ABS had the same conclusion [55,56,57]. Table 5 presents the statistical data regarding the influence of temperature. The mean data was calculated by all specimens with the same temperatures regardless of other parameter settings. The subsequent statistical analysis of other parameters was also based on the same processing. By increasing the environmental temperature from 50 to 70 °C, the mean number of cycles until failure drops from 96,545 to 31,774 drastically.

This most significant influence of temperature on fatigue life was due to the deterioration of the mechanical properties with the increase in temperature. The FDM ABS appeared to be glassy at the lower temperatures in terms of the molecular microstructure. In other words, the rotation of the molecular chain ceased to occur and, therefore, the coil could not uncoil and lengthen [58]. The side group and the chain elements could move in a small area. By increasing the environmental temperature, the energy of the moving units and movable space increased. Additionally, the van der Waals force between molecules decreased. The chemical bonds between the molecules were easy to break, and as such, the chain slip occurred easily [59]. Therefore, it can be concluded that microstructural fractures, crack initiation, and propagations are easier in FDM ABS at higher temperatures. Furthermore, such changes in the microstructural properties influence the mechanical behaviour of the specimens. Finally, both the change in the microstructural and mechanical behaviour in FDM ABS contribute significantly to the fatigue life of the beam. The change in the mechanical behaviour is further evidenced from the DMA results presented later herein.

#### 3.2.3. Building Orientation Influence

The building orientation is the most critical printing parameter affecting the fatigue life of the FDM ABS beam. Figure 12 show the variation in the fatigue life for different building orientations. The specimens with the X building orientation have the highest number of cycles in all the subplots. The Y building orientation provides the worst fatigue performance, whereas the performance provided by the XY building orientation lies between them. Table 6 lists the average fatigue life for groups with different building orientations. The X building orientation provides the longest fatigue life with an average of 96,545 cycles until fracture. However, the number of cycles is only 40,699 for beams with the Y building orientation. It is less than half the fatigue life of the X orientation. The experimental results are different with those presented in most previous studies, which performed the tensile fatigue test and reported that X orientation had the best fatigue strength [10,28,29,34,35,36]. The difference between tension and bending fatigue tests may cause this different result.

The results are reasonable. The initial seeded crack is lateral on the beam, the same as that in the case of the specimen with a Y building orientation. In fracture mechanics, the crack initiates and propagates from micro-cracks on or in the structure [60]. These micro-cracks can be represented by 3D printing defects in the research. The micro-voids occur between the filaments in the structure. The presence of these air voids between the fibres leads to stress concentration when the beam vibrates.

Furthermore, the bonds between the filaments are the weakest in the areas where the voids are present. The occurrence of crazing is earlier in the area with the voids. Notably, the direction of the void area is the same as that of the initial-seeded crack. This provides an excellent crack path that leads to a quick crack propagation. On the contrary, due to the beam’s vibration, the bending stress is longitudinal and acts on the Y orientation voids vertically, thereby accelerating the crack growth and decreasing the fatigue life.

#### 3.2.4. Nozzle Size Influence

Figure 13 show the variation tendency of fatigue life for different nozzle sizes. As can be seen, the fatigue strength has an increasing trend when the nozzle size increases for all the specimens. The tests had the similar conclusion with FDM PLA [39,40,41]. Table 7 present the mean fatigue life for the specimen group with different nozzle sizes. The mean number of cycles increases from 45,195 for the 0.4-mm nozzle size to 84,087 for the 0.8 mm nozzle size. The nozzle size also significantly influences the manufacturing time.

The effect of nozzle size on the fatigue life was similar to that in the case of a previously reported micro air voids theory. We assumed the spacing between the void defects during the printing process to be almost constant between the filaments, regardless of the thickness of the filament. This implies that the beam printed with a 0.8 mm-nozzle had fewer micro-voids than that printed with a 0.4-mm nozzle. Fewer void defects increase the global density of the specimen, thus reducing the stress concentration. The specimen then has a higher storage modulus, and the crack propagation becomes considerably difficult. Therefore, the overall fatigue life increases with the increase in the size of the nozzle.

#### 3.2.5. Influence of Layer Thickness

We can verify the results obtained from the analysis of variance (ANOVA) test in Section 3.2.1 based on the data presented in Figure 14. The fatigue life only increases slightly when the layer thickness changes from 0.05 mm to 0.15 mm for all the specimens. A similar trend was reported in FDM PLA research [39,40,41]. Despite the ANOVA results regarding the layer thickness, the influence of layer thickness on the average fatigue life is not entirely clear based on the presented data. The statistical results are presented in Table 8 demonstrate the potential relationship between layer thickness and fracture resistance. The average fatigue life of the beam with a 0.15-mm layer thickness (74,463 cycles) is slightly higher than 0.1-mm (65,196 cycles) and 0.05-mm (53,969) layer thicknesses.

The possible cause of the effect of layer thickness may be the same as that in the case of nozzle size. Since micro air voids exist between each layer, the size of the voids is also almost constant and does not relate to the layer thickness. As a result, the beam with a thicker layer has fewer microvoids. Therefore, a 0.15-mm thick layer provides the best fatigue performance. 

### 3.3. Technological Recommendations for 3D Printing

The results obtained for fatigue life, as presented above, can help us obtain the optimal 3D printing parameter combination to ensure a better bending fatigue life within a certain parameter range investigated in this research. These parameters are listed in Table 9. A nozzle size of 0.8 mm and a layer thickness of 0.15 mm also had a significant impact on the fatigue life as the use of these parameters led to a significant reduction in the manufacturing time of the parts.

### 3.4. Fractography

Different pictures of the specimen fracture section were taken with a Dino-Lite digital microscope (AnMo Electronics Corporation, Hsinchu, China) after the bending fatigue tests. Figure 15 shows which surface is captured by the microscope. All the pictures in the following figures show the different levels of stress-whitening. Furthermore, it can be seen that the mixture of ductile and brittle break occurs during the fracture for all the specimens. The white area represents the ductile fracture, but the red area represents the brittle fracture. 

As a universal phenomenon in polymers, stress-whitening is caused by microvoids and crazes [61]. The polymer chains reorganize during tension when the beam is vibrating. The straightening, slipping, and shearing of the fillers and the polymer chains during movement within the plastic’s microenvironment may lead to the formation of occlusions or holes in some cases. When these occlusions are combined, microvoids are produced. The transmitted light is scattered when the microvoids cluster to a size greater than or equal to the wavelength of light (380–750 nm). The microvoids change the refractive index of the plastic, causing the object to appear white.

In other words, the defects (voids) of the 3D printing ABS cause local stress concentration and these areas are more likely to induce crazes. Figure 16, Figure 17 and Figure 18 show the specimens with different nozzle sizes. Similarly, printing defects occur between each filament. However, the distance between each void, representing the nozzle size, varies in the horizontal direction. As can be seen from the aforementioned figures, all the pictures show some bright white lines extending from the printing voids along the direction of crack growth, confirming that the crack propagates along the microvoids and crazes. Then, the void numbers in the cross-section of specimens were calculated approximately. For the 0.15 mm layer thickness specimen, the 0.4 mm nozzle printed a total 500 air voids. This was more than the 333 voids of 0.6 mm and 250 voids of 0.4 mm. This can also be visually observed from the Figure 16, Figure 17 and Figure 18. Because the number of voids within the same area cross-section of a 0.4-mm nozzle (61) is more than that of a 0.6 (40) and 0.8 mm nozzle (31), consequently, the number of stress whitening areas is also greater in the former. This results in a higher fatigue crack growth rate and short fatigue life.

Furthermore, from the figures mentioned above, it can be observed that all the pictures show that the crack growth rate is time dependent. The white and red areas alternately appear along the crack propagation direction. This means the crack growth dominated by the ductile break and brittle break alternately occur. The layer-by-layer 3D-printed structure may lead to the phenomenon. Furthermore, the brittle break rate in the red area is higher than the ductile break in the white area. It is also reflected on the global colour distribution along the crack propagation direction on the section. Colour near the beam surface is significantly whiter than the bottom area. The ductile break gradually changes to a brittle break during crack propagation. This implies that the initial crack growth rate is less than the latter because the crack tip’s stress may increase along with the depth of the crack. The greater the stress, the faster and easier the brittle fracture.

Figure 19 show the cross-sectional area for the XY building orientation. Compared to Figure 16, Figure 19 shows a larger area of voids. Furthermore, the larger stress whitening area connects the voids. These larger area voids decreased the strength of a structure which led to the decreased fatigue life than X building orientation. However, from Figure 20, which presents the fracture section of the Y printing orientation, we can see that there is almost no stress whitening in this section. This is because this section has the weakest bonds between the two filaments. Therefore, the brittle fracture continues along this section. The voids between each filament and layer, represented by dark red, occupy a large area of the fracture surface, thereby speeding up the crack growth rate.

The inter-filament bonding strength along the crack path in the specimen with a Y building orientation is also extremely weak. As can be seen from Figure 20 and Figure 21, some loose filaments are peeled off the fractured surface.

For the layer thickness effect, when Figure 22 and Figure 23 are compared, we can see that the section with a 0.15 mm layer thickness has fewer defects (dark red area) between the layers in the unit section. This may be the reason why it has a longer fatigue life. 

## 4. Conclusions

The influence of printing parameters (building orientation, nozzle size, and layer thickness) and environmental temperature on the bending fatigue life of a FDM ABS beam was investigated. The following conclusions were drawn:The environmental temperature has the greatest influence on the fatigue performance, followed by the building orientation and nozzle size, whereas the layer thickness does not show a significant influence with a 0.164 *p*-value of ANOVA;A combination of the following parameters provides the longest fatigue life among the tested values: X building orientation, 0.8 mm nozzle size, and 0.15 mm-thick layer;Higher temperature reduces the fatigue life possibly due to more active molecular movement. The mean fatigue life of beams under 70 °C is approximate 32,000 cycles. It is only one third of that under 50 °C.Parts with a Y building orientation have the lowest mean fatigue life (around 41,000 number of cycles), compared with 86,000 number of cycles of X building orientation. The reason is that the micro air voids in a Y building orientation beam have the same direction as the initial-seeded transverse crack. This reduces the strength of the potential fracture surface.Both a larger nozzle size and thicker layer height decrease the beams’ micro void space and quantity per unit area in the potential crack path and lead to a higher fatigue resistance.

It was found that printing void defects fundamentally affect the fatigue life of the FDM structure. Future work will extend the analytical modelling of the relationship between these printing parameters and the bending fatigue life of the ABS beam.

## Figures and Tables

**Figure 1 polymers-13-02362-f001:**
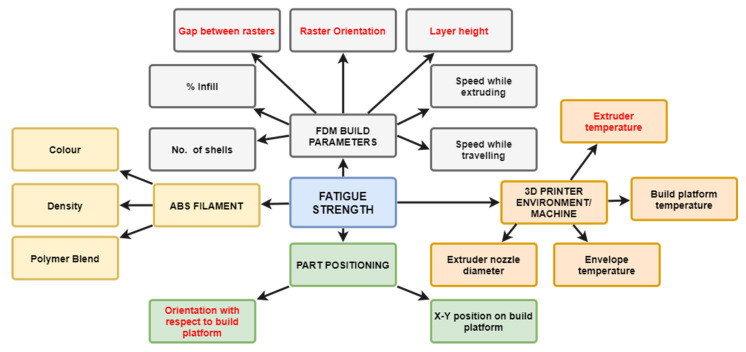
Critical parameters relating to the fatigue performance of the fused deposition modelling (FDM) structure [27].

**Figure 2 polymers-13-02362-f002:**
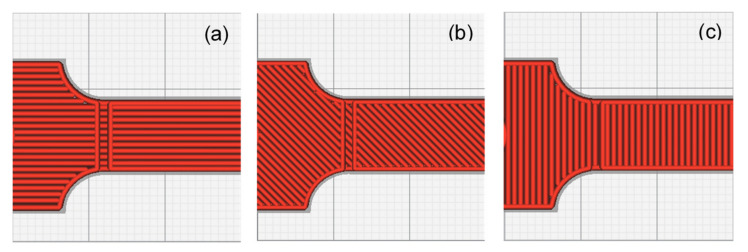
Printing directions: (**a**) X direction; (**b**) XY direction; (**c**) Y direction.

**Figure 3 polymers-13-02362-f003:**

Nozzle size: (**a**) 0.4 mm; (**b**) 0.6 mm; (**c**) 0.8 mm.

**Figure 4 polymers-13-02362-f004:**
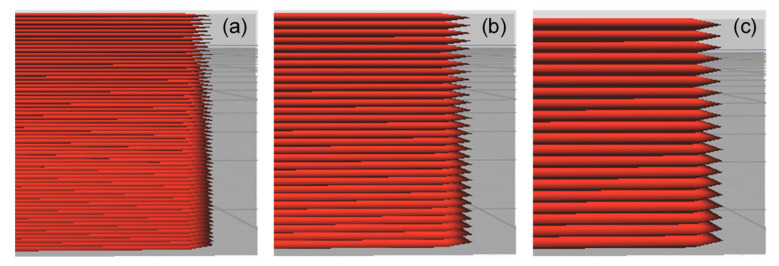
Layer thickness: (**a**) 0.05 mm (**b**) 0.10 mm (**c**) 0.15 mm.

**Figure 5 polymers-13-02362-f005:**
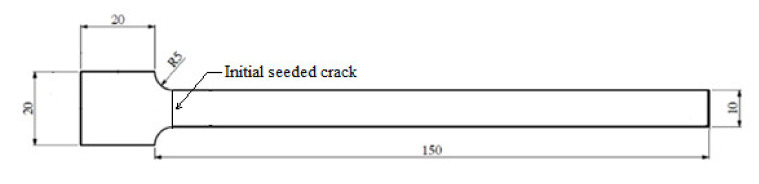
Geometry of specimen.

**Figure 6 polymers-13-02362-f006:**
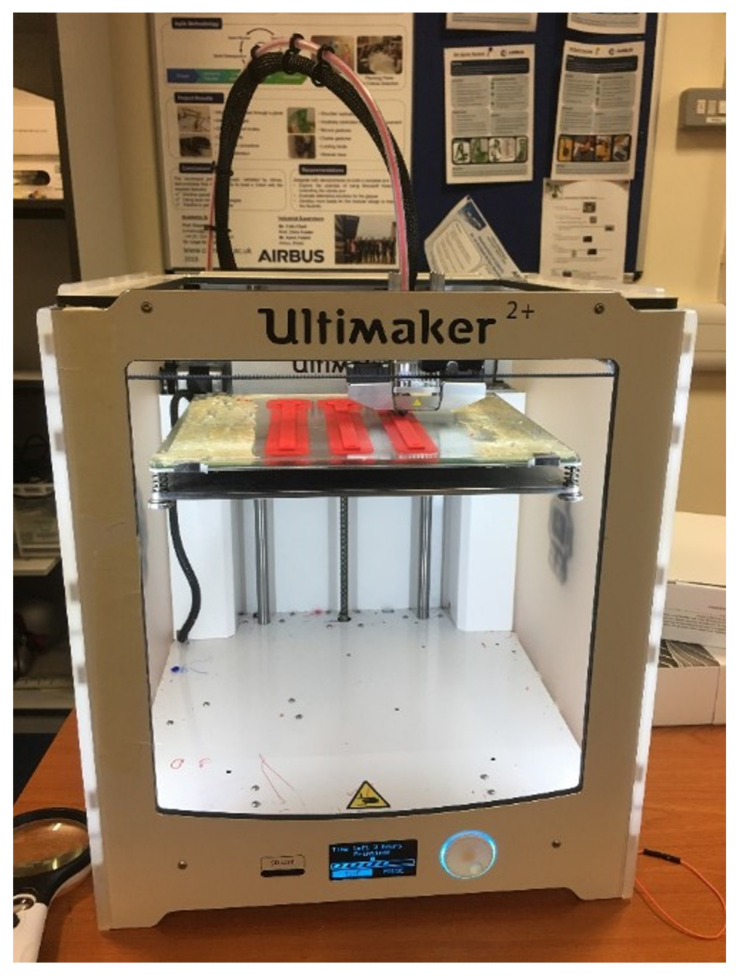
Three-dimensional (3D) printing by Ultimaker 2+ printer.

**Figure 7 polymers-13-02362-f007:**
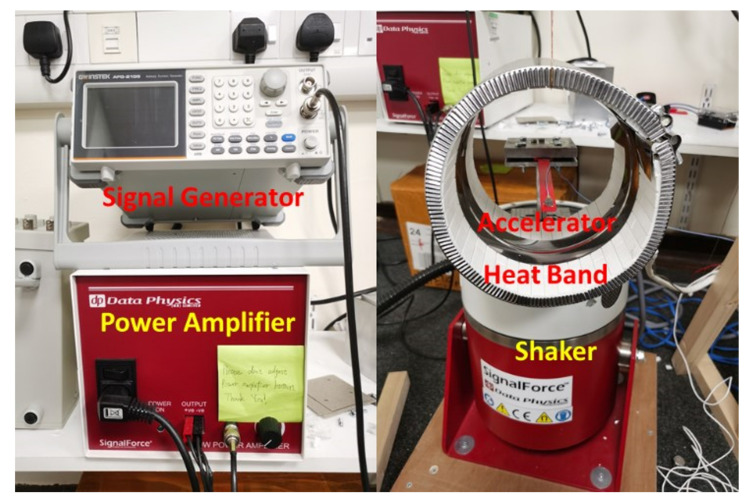
Experimental setup.

**Figure 8 polymers-13-02362-f008:**
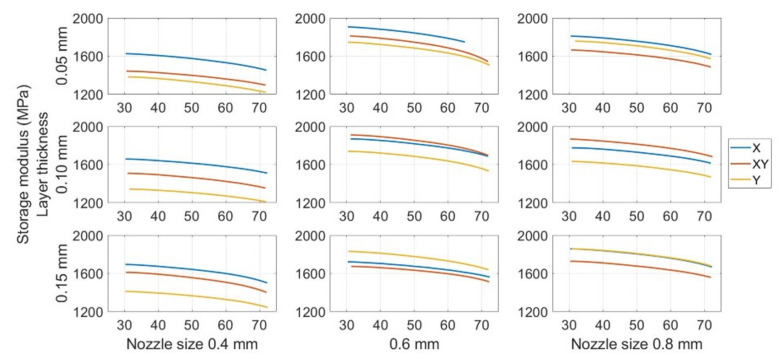
Detailed storage modulus change for different printing parameters from 30 to 70 °C.

**Figure 9 polymers-13-02362-f009:**
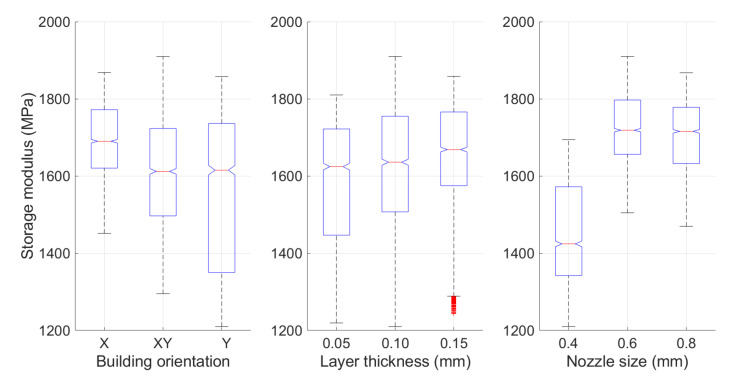
Statistical analysis of the storage modulus for different printing parameters.

**Figure 10 polymers-13-02362-f010:**
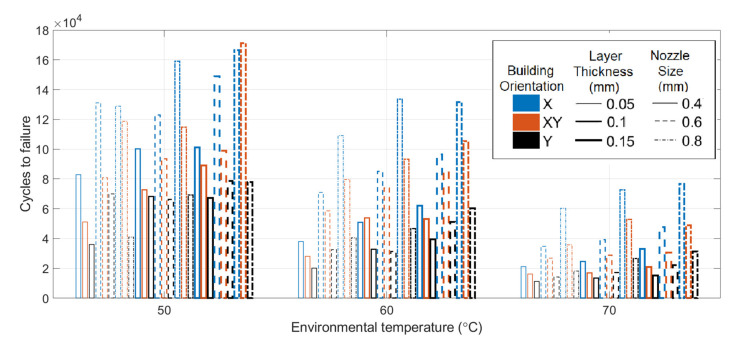
Mean number of cycles of specimens at different temperatures.

**Figure 11 polymers-13-02362-f011:**
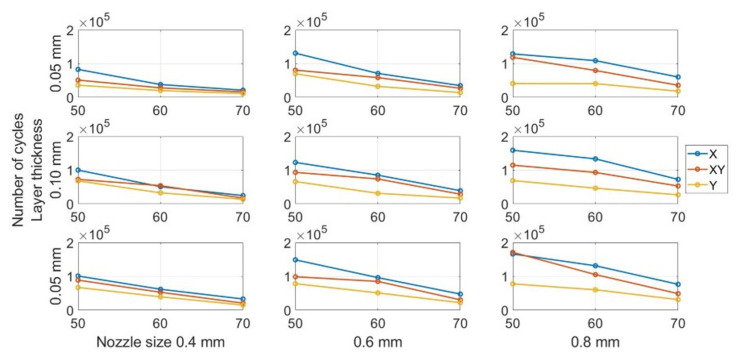
Effect of temperature on fatigue life.

**Figure 12 polymers-13-02362-f012:**
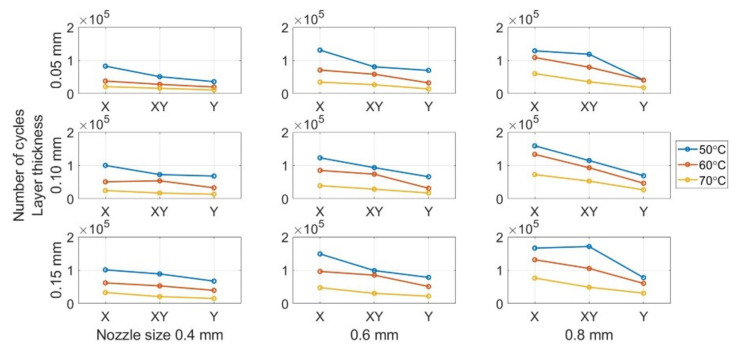
Building orientation effect on fatigue life.

**Figure 13 polymers-13-02362-f013:**
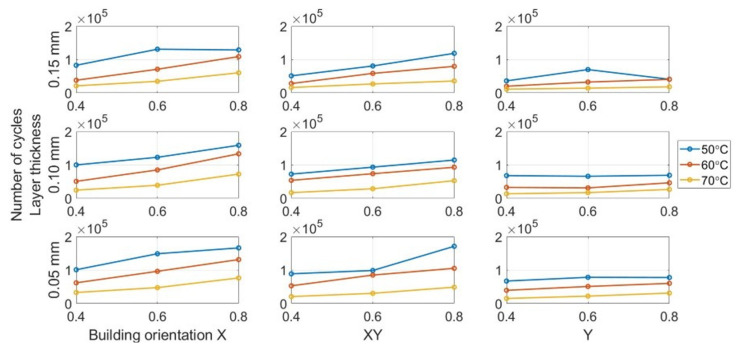
Influence of nozzle size on fatigue life.

**Figure 14 polymers-13-02362-f014:**
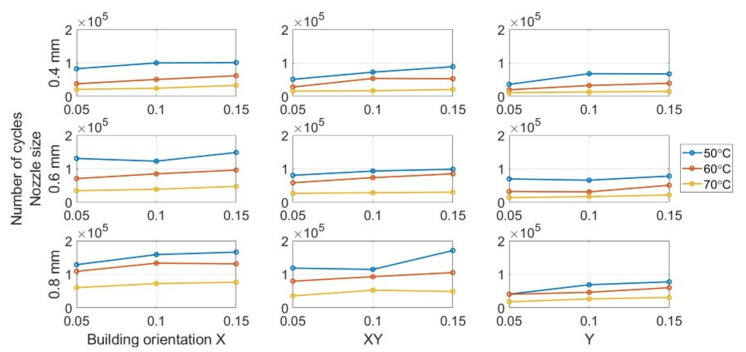
Layer thickness influence on fatigue life.

**Figure 15 polymers-13-02362-f015:**
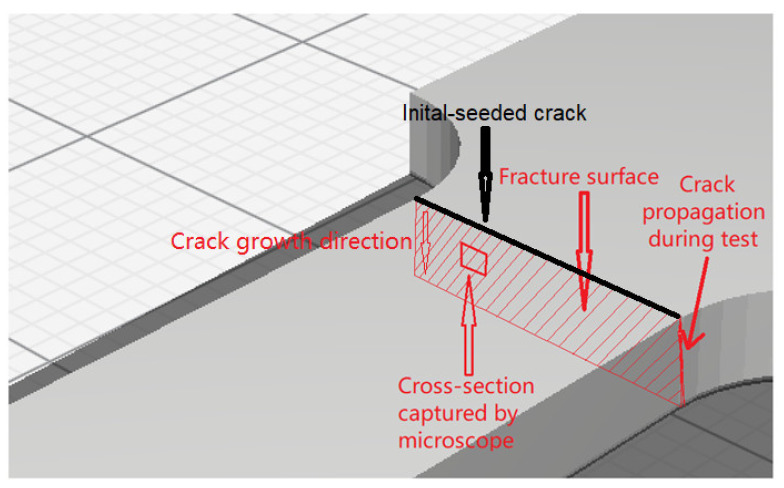
A schematic to show which cross-section was captured by the Dino-Lite digital microscope.

**Figure 16 polymers-13-02362-f016:**
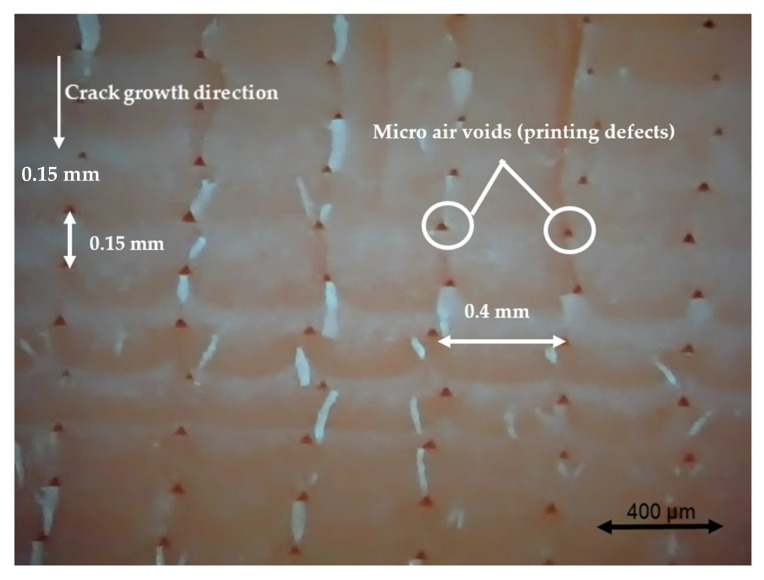
Specimen with a nozzle size of 0.4 mm, X building orientation, and layer thickness of 0.15 mm.

**Figure 17 polymers-13-02362-f017:**
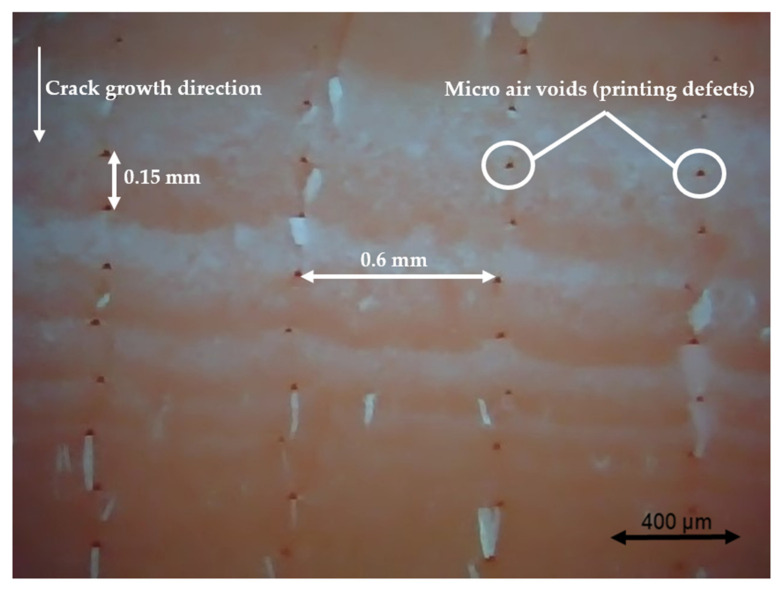
Specimen with a nozzle size of 0.6 mm, X building orientation, and layer thickness of 0.15 mm.

**Figure 18 polymers-13-02362-f018:**
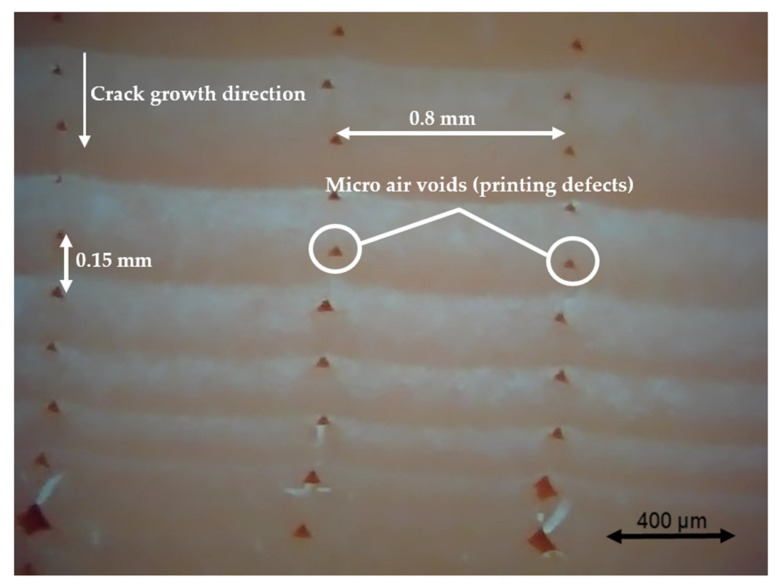
Specimen with a nozzle size of 0.8 mm, X building orientation, and a layer thickness of 0.15 mm.

**Figure 19 polymers-13-02362-f019:**
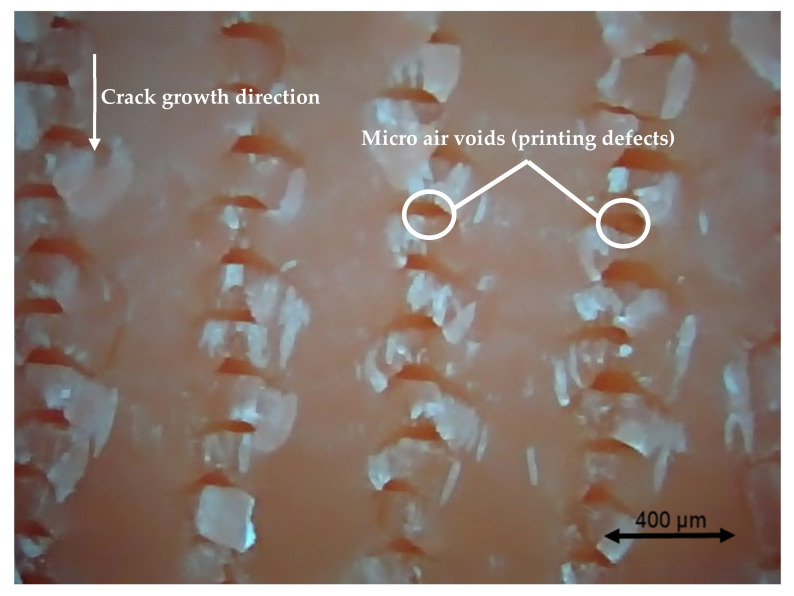
Specimen with nozzle sizes of 0.4 mm, XY building orientation and layer thickness of 0.15 mm.

**Figure 20 polymers-13-02362-f020:**
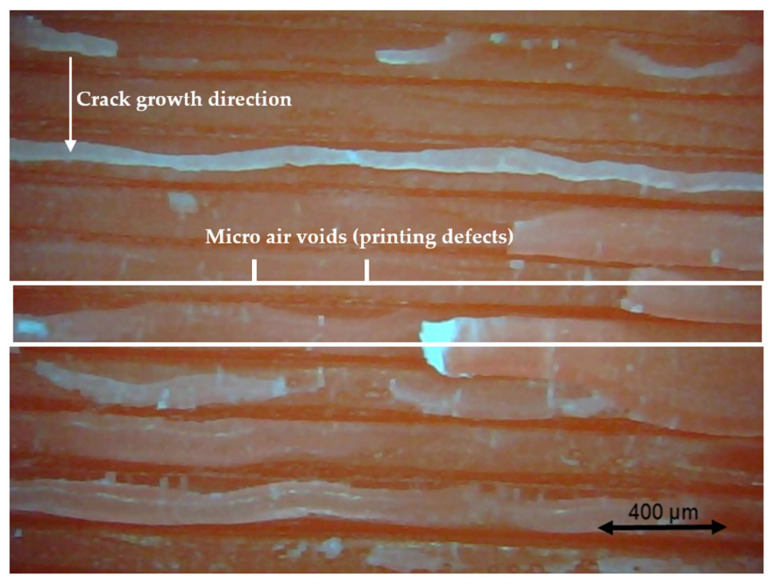
Specimen with a nozzle size of 0.4 mm, Y building orientation, and layer thickness of 0.15 mm.

**Figure 21 polymers-13-02362-f021:**
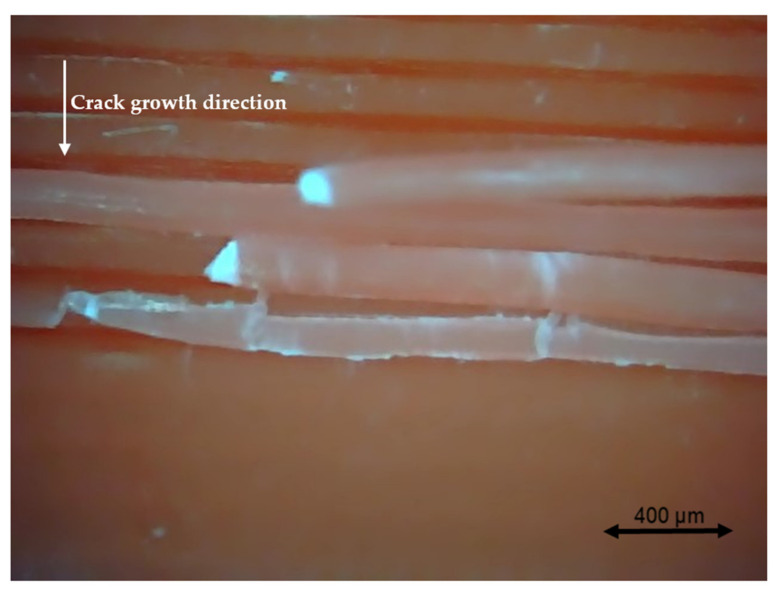
Specimen with a nozzle size of 0.8 mm, Y building orientation, and layer thickness of 0.15 mm.

**Figure 22 polymers-13-02362-f022:**
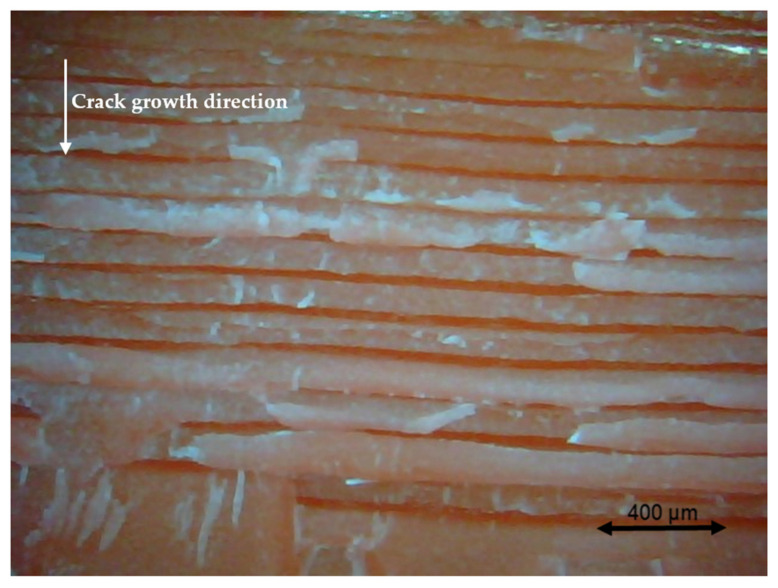
Specimen with a nozzle size of 0.6 mm, Y building orientation, and layer thickness of 0.1 mm.

**Figure 23 polymers-13-02362-f023:**
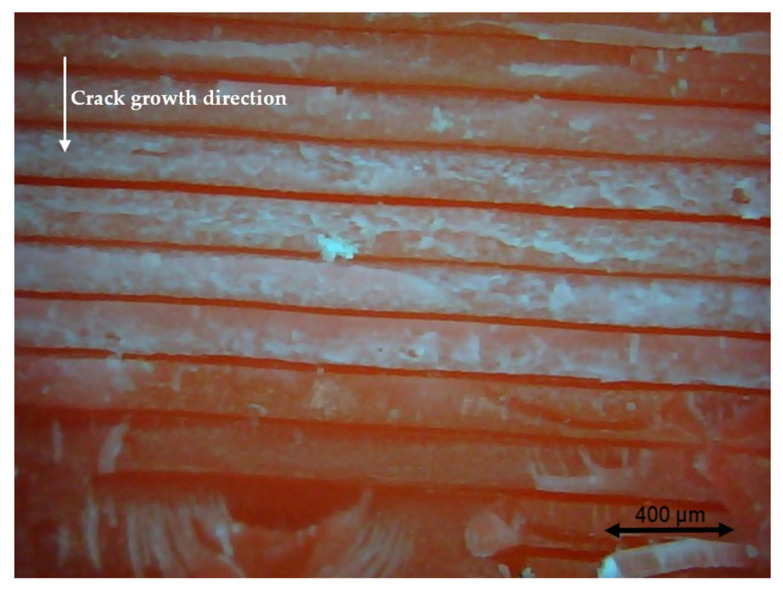
Specimen with a nozzle size of 0.6 mm, Y building orientation, and layer thickness of 0.15 mm.

**Table 1 polymers-13-02362-t001:** Printing parameters.

Building Orientations	Nozzle Size (mm)	Layer Thickness (mm)
0° (X)	0.4	0.05
±45° (XY)	0.6	0.10
90° (Y)	0.8	0.15

**Table 2 polymers-13-02362-t002:** Ultimaker^®^ acrylonitrile butadiene styrene (ABS) filament specifications [45].

Filament Specification	Value
Diameter	2.85 ± 0.10 mm
Tensile modulus	1618.5 MPa (ISO 527) [46]
Elongation at yield	3.5% (ISO 527)
Flexural modulus	2070 MPa (ISO 178) [47]
Vicat softening temperature	97 °C
Melting temperature	225–245 °C

**Table 3 polymers-13-02362-t003:** Mean storage modulus for different printing parameters.

Printing Parameters	Group	Mean Storage Modulus (MPa)	*p*-Value
Building Orientation	X	1692	2.34 × 10^−115^
	XY	1616	
	Y	1561	
Nozzle Size (mm)	0.4	1450	0
	0.6	1722	
	0.8	1708	
Layer Thickness (mm)	0.05	1590	2.30 × 10^−31^
	0.10	1622	
	0.15	1648	

**Table 4 polymers-13-02362-t004:** Statistical parameter values.

Parameter	Chi-Square	*p*-Value
Building Orientation	17.79	0.0001
Nozzle Size	11.81	0.0027
Layer Thickness	3.56	0.1684
Temperature	41.24	1.11 × 10^−9^

**Table 5 polymers-13-02362-t005:** Mean number of cycles until fracture at different temperatures.

Environmental Temperature (°C)	Mean Number of Cycles until the Fracture	Standard Deviation
50	96,545	36,969
60	65,312	31,163
70	31,773	17,747

**Table 6 polymers-13-02362-t006:** Mean number of cycles until fracture for different building orientations.

Building Orientation	Mean Number of Cycles until the Fracture	Standard Deviation
X	86,270	43,307
XY	66,659	37,338
Y	40,699	21,850

**Table 7 polymers-13-02362-t007:** Mean number of cycles until fracture for different nozzle sizes.

Nozzle Size (mm)	Mean Number of Cycles until the Fracture	Standard Deviation
0.4	45,195	27,329
0.6	64,346	36,201
0.8	84,087	44,604

**Table 8 polymers-13-02362-t008:** Mean number of cycles until fracture for different layer thickness.

Layer Thickness (mm)	Mean Number of Cycles until the Fracture	Standard Deviation
0.05	53,969	35,857
0.10	65,196	38,314
0.15	74,463	43,126

**Table 9 polymers-13-02362-t009:** Optimal printing parameter combination for a longer fatigue life.

Parameter	Optimal Level for Greater Resistance to Fatigue
Building Orientation	X
Nozzle Size	0.8 mm (The maximum nozzle provided by Ultimaker^®^)
Layer Thickness	0.15 mm

## Data Availability

The data presented in this study are available on request from the corresponding author.

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
