# Peer review of "Effects of Printing Parameters on the Fatigue Behaviour of 3D-Printed ABS under Dynamic Thermo-Mechanical Loads"

_polymers, 2021, doi:10.3390/polym13142362_

Round 1
Reviewer 1 Report
The authors have updated their manuscript and attempted to respond to prior comments. However, several of the previous comments were not actually addressed. Unfortunately, there remains many references to literature that do not contain citations. For instance, on page 2 line 54 “However, only a few studies…” and page 2 line 76 “Compared with the prior research….” neither are not followed with any citations. Only what studies? Compared to what research? By whom? There are many similar instances. The authors need to do a careful reading of their manuscript and add relevant citations where noted and necessary.
Reply:
The authors all agree that a large amount of previous research investigated the fatigue behaviour of several FDM materials. However, as the author mentioned in the manuscript, the previous research all carried out the tensile fatigue test with the pure mechanical loads. This paper applied the flexural fatigue test with the dynamic thermo-mechanical loads. As for the listed reference paper, the manuscript has included them.
The authors would then be encouraged to evaluate literature for comparison of the flexural fatigue, which has also been performed. For instance, https://doi.org/10.1016/j.matdes.2018.06.018, among others are found with simple searches.. The authors should perform more due diligence.
Comment:
- The authors report storage modulus to the 0.1 MPa (1691.1 MPa). Is that a meaningful level of precision for the study. No. The authors should revisit their statistical analysis and reconsider how they present their quantitative results. This is especially true given the size of the distribution and error bars on Figure 9.
Reply:
Authors explained the data shown in box plot and used in statistical analysis. There is no error bar in Figure 9.
I feel the authors missed the point on this prior comment. Regardless of what a statistical analysis software reports, the data must be presented to the level of precision of the experiments. For example, many students do calculations but wouldn’t dare to report the 13 decimals their calculator tells them. So, do the authors believe that their mean storage modulus in Table 3 is actually meaningfully accurate to 5 significant digits and the 0.1 MPa? I do not believe that is meaningful. The authors appear to agree as in text they make notes like “over 1700 MPa” and “the 1450 MPa mean”. Why not write the 1450.4 MPa, if the authors believe that decimal is meaningful?
Figure 10 remains difficult to parse.
Pg 15 line 453: what does “The bonds strength along the crack path..” mean? Surely not the polymer chain bonds. Revise.
The discussion on Figures 15-21 is very weak to non existent.
Reviewer 2 Report
The authors have resubmitted the manuscript and indicated changes, in the manuscript. The authors have conducted revisions but have not fully addressed all concerns.
The authors are vague in explaining how they indicated some changes in the manuscript without providing specific points in the manuscript for where they took place. The comments in the manuscript make it difficult to read once printed because all of the font is smaller to accommodate the comments with many superfluous comments such as reformatting. Providing an additional version with font color changed to indicate revisions with no comments would improve readability for reviewing.
Here is a list of comments in the previous manuscript these are not fully addressed, with comments for this round in quotes:
Comment:
The topic is of interest, but the authors have not clearly communicated why fatigue is important to measure in comparison to other mechanical properties for 3D printed parts and how it could affect applications in the future.
Reply:
Authors revised and introduce the importance and influence of fatigue in introduction part.
“It is unclear what thermo-mechanical loading conditions are and why they are important to study, the description of practical working conditions is vague.”
Comment:
By what metric/evidence is ABS considered the most commonly used plastic?
Reply:
Authors cited the reference about this conclusion.
“There should not be a reference in the abstract, perhaps stating “is one of the most commonly” used plastics is acceptable.”
Comment:
Many parameters are listed in Figure 1, but why are a few highlighted in red and focused on for this study?
Reply:
Authors revised it and explained the meaning of colour difference. Red showed the most likely dominant
factors.
“It is not clear the criteria used to determine why these are the most dominant factors, and what are they dominant for? Thermo-mechanical loading properties?”
Comment:
Generally, there is little comparisons to other studies in the introduction and no outline for what is expected in the paper.
Reply:
The main difference of the research and prior studies are two points: the extra thermo loads and dynamic
bending stress. They make the experimental environmental closer to practical working conditions. Author
also mentioned the expectation in revised manuscript.
“This comment is partially addressed, an outline or set of steps that will be conducted throughout the study would still improve the closing of the introduction.”
Comment:
On Page 3 it is unclear why different environments for thermal load are considered in addition to print
processing parameters.
Reply:
Authors explained the significance of environmental temperature in revised manuscript.
“This comment has not been fully addressed, why these temperatures? What is the environment representing?”
Comment:
For Table 3 it is not clear what samples are being tested and what the p-value is comparing. It is not clear
why storage modulus is measured and its relation to fatigue.
Reply:
Authors introduced the samples and p-value in Section 3.1. Author also revised and explained the .
Comment:
“The presented p-values are still odd, why is it 0 for nozzle size and seems incredibly small for the other parameters, how accurate are these to that many decimal places? What do we learn from this statistical comparison, it seems obvious at least one change in the group would be different. How accurate are the mean storage modulus values in terms of uncertainty for calculating these p-values?”
P9 lines 239 need to be cited and further qualified to support the claim of molecular chain rotation ceasing, including subsequent text in the section.
Reply:
Authors cited the reference.
“Is this for citation [44]? It is not clear that citation 44 is referring to the molecular chain rotation.”
Comment:
P10 there is no direct evidence demonstrated for micro-cracks, this needs to be demonstrated with
microscopy and otherwise cited otherwise these explanations are not justified to interpret the data at this
point in the results.
Reply:
Authors cited the reference and revised the sentence to make sense. The 3D printed defects play the role of micro-crack in fracture mechanics.
“It is not clear what change occurred to address this comment. Overall microscopy images are confusing to interpret and link back to how the microscopy represents a cross-section of the original structure. This should be better explained with perhaps a reference figure. It is not clear why there are elongated bright white spots on the sample, is this light reflecting? Crack growth direction is highlighted, but it is not clear where cracks are in these images.”
Comment:
When aggregating data for Table 6, which samples are included? This is confusing for a number of points
in the paper, are all samples averaged with the same building orientation regardless of other parameter
settings?
Reply:
Yes. We revised it with more clear expression.
“It is not clear what change occurred to address this comment.”
Comment:
In section 3.3 these can not be concluded as the optimal printing parameters, the 0.8mm and 0.15mm
represent extremes in the experiments so it is unclear if further increasing these values would better improve fatigue. Additionally, there is the potential for better combinations of printing parameters in between
parameter levels investigated.
Reply:
Authors totally agree with the comment. We revised the sentence for more accurate expression in
manuscript. The research has listed all potential parameter combinations so we believed the 0.8mm and
0.15mm were the optimal parameter in the range covered by the research.
“It is not clear what change occurred to address this comment.”
Comment:
From pages 13 onwards the microscopy images are poorly labelled so it is unclear what authors are
specifically referring to from the text in relation to interpreting the images. Specific cracks/features of
interest should be labelled on the image. The last few microscopy samples are poorly explained/integrated
with the text and there are no images demonstrating fractures/failures of the part.
Reply:
All pictures are the fracture section of specimens. Authors revised them and labelled crack propagation
direction, micro-air voids (printing defect), etc. Figure 20 and 21 showed the different sections for different
layer thickness. Authors explained them in manuscript.
“The labelling is improved, but in-text descriptions could still use further explanation to clarify how interpreting and comparing the microscopy leads to a fuller understanding of fatigue behavior. For instance, figure 17 is confusing because of the image quality it is not clear where on the sample is being imaged and layer/cracks are not clearly visible.”
Comment:
The authors use qualitative words such as “slight impact” whereas quantitative measures would provide
better comparisons to justify claims.
Reply:
Authors revised the expression in conclusion part.
“If the p-value is greater than 0.1, how are the author’s assessing impact? Typically this is not considered significant.”
Comment:
Some conclusions, such as conclusion 3 was not directly supported by the study or cited to justify the claim and other possible interpretations of the data.
Reply:
Authors recorded the reduced fatigue life under increased temperature during the tests. As for the possible reason, authors revised and cited the reference in Section 3.2.2 to support the conclusion.
“Since this is not directly observed in the paper, it is recommended to refer as “possibly due to more active molecular movement.””
Comment:
When referring to printing voids, was there any quantification of the number/size of printing voids in
different parts to justify claims of their affects on trends?
Reply:
Thank you for the suggestion. Authors revised and counted the number of printing voids in Section 3.4
“There is no revision to indicate a potential difference in size between the sample, for the counting is this based on the area viewed in the microscope slide? It could potentially be calculated based on the layer size and deposition pattern.”

Round 2
Reviewer 1 Report
Prior Comment 1: The authors all agree that a large amount of previous research investigated the fatigue behaviour of several FDM materials. However, as the author mentioned in the manuscript, the previous research all carried out the tensile fatigue test with the pure mechanical loads. This paper applied the flexural fatigue test with the dynamic thermo-mechanical loads. As for the listed reference paper, the manuscript has included them.
The authors would then be encouraged to evaluate literature for comparison of the flexural fatigue, which has also been performed. For instance, https://doi.org/10.1016/j.matdes.2018.06.018, among others are found with simple searches. The authors should perform more due diligence.
Reply:
Authors used more rigorous expression in revised manuscript. The literature review was carried out focusing on the FDM ABS material. No bending/flexural fatigue test was done for FDM ABS structure. Authors cited the flexural fatigue literature mentioned in the comment and compared works.
New Review Comment: I find that to be insufficient. The work needs to be put in context of the findings for the same behavior of other materials. Simply because it hasn’t been done on polymer X, doesn’t mean you can ignore how polymer Y behaved. Are there differences? Are there similarities? Why or why not? Findings for other materials should be noted in the intro (if it’s interesting it should motivate your study!) and used in the discussion of the authors results for ABS.
Comment:
- The authors report storage modulus to the 0.1 MPa (1691.1 MPa). Is that a meaningful level of precision for the study. No. The authors should revisit their statistical analysis and reconsider how they present their quantitative results. This is especially true given the size of the distribution and error bars on Figure 9.
I feel the authors missed the point on this prior comment. Regardless of what a statistical analysis software reports, the data must be presented to the level of precision of the experiments. For example, many students do calculations but wouldn’t dare to report the 13 decimals their calculator tells them. So, do the authors believe that their mean storage modulus in Table 3 is actually meaningfully accurate to 5 significant digits and the 0.1 MPa? I do not believe that is meaningful. The authors appear to agree as in text they make notes like “over 1700 MPa” and “the 1450 MPa mean”. Why not write the 1450.4 MPa, if the authors believe that decimal is meaningful?
Reply:
Authors agree with the comment and change the level of precision.
New Review Comment: The authors missed a couple instances. For example, on pg. 8 0.1MPa precision is still used. Revise thoroughly.
Reviewer 2 Report
The authors have provided revisions and addressed most concerns, although the presented revisions are still not very friendly for a reviewer to check. In the future the authors should more clearly indicate where changes have been made and what changes have been made in the response document rather than vaguely referring the reviewer to check the main manuscript.
The concerns that need to be addressed prior to publishing, it is still not clear why the environmental temperature was changed for different tests and what real-world conditions may be relevant.
The p-values and items in many of the tables are not reported with an appropriate number of significant digits given the uncertainties in the experiments, in addition to other data such as reported standard deviations.
In the microscopy images they are still not high quality but sufficient for publication if explained what some of the artifacts in the images are (such as the white periodic shines throughout) and whether the crack growth direction is the same every time from the same point, how is this determined? This is also not clear in the added schematic.
Author Response
Please see the attachment.

This manuscript is a resubmission of an earlier submission. The following is a list of the peer review reports and author responses from that submission.
Round 1
Reviewer 1 Report
file attached.

Reviewer 2 Report
The paper investigates how printing parameters influence loading before failure for fatigue testing of 3D printed parts. Overall the experiments seem carried out carefully, but lack justification in the writing and interpretation so the paper can not be recommended for publication as submitted. The topic is of interest, but the authors have not clearly communicated why fatigue is important to measure in comparison to other mechanical properties for 3D printed parts and how it could affect applications in the future.
Some specific comments are as follows:
Abstract:
There should be no “and” in the first sentence before “effective.”
“Several” in the second sentence of abstract should be changed, there are many more than several types of polymers used in FDM.
By what metric/evidence is ABS considered the most commonly used plastic?
The abstract is vague in that it doesn’t specify any trends of fatigue performance in relation to design parameters or the value of measured fatigue.
Introduction:
Authors use oxford comma during abstract but in line 38 on page 1 do not, this should be consistent throughout the manuscript.
In page 1 line 44, to what extent is FDM ABS reduced in mechanical properties?
In page 1 line 44 the references are not combined correctly for 14 and 15.
The introduction is very underdeveloped, Figure 1 is not explained fully and it is unclear why it is important to study fatigue behavior in 3D printed parts. The argument that few studies have investigated a topic such as bending fatigue is not sufficient to motivate studying the topic if it is unimportant, the authors require further motivation for their study and description of their specific hypotheses and experimental design.
Many parameters are listed in Figure 1, but why are a few highlighted in red and focused on for this study?
Generally, there is little comparisons to other studies in the introduction and no outline for what is expected in the paper.
Methodology:
On Page 3 it is unclear why different environments for thermal load are considered in addition to print processing parameters.
The authors should propose hypothesis or rationale for how these parameter changes could affect fatigue in the introduction.
On page 4 line 103 it is not justified why crack growth and plastic zones are observed in the chosen material but not others.
It is not cited or explained where Table 2 values are obtained.
By the end of the methodsit is not clear how many samples were tested and how many parameter combinations were tested, were all 81 combinations tested multiple times to achieve statistical significance?
Results
For Table 3 it is not clear what samples are being tested and what the p-value is comparing. It is not clear why storage modulus is measured and its relation to fatigue.
P9 lines 239 need to be cited and further qualified to support the claim of molecular chain rotation ceasing, including subsequent text in the section.
P10 there is no direct evidence demonstrated for micro-cracks, this needs to be demonstrated with microscopy and otherwise cited otherwise these explanations are not justified to interpret the data at this point in the results.
When aggregating data for Table 6, which samples are included? This is confusing for a number of points in the paper, are all samples averaged with the same building orientation regardless of other parameter settings?
In section 3.3 these can not be concluded as the optimal printing parameters, the 0.8mm and 0.15mm represent extremes in the experiments so it is unclear if further increasing these values would better improve fatigue. Additionally, there is the potential for better combinations of printing parameters in between parameter levels investigated.
From pages 13 onwards the microscopy images are poorly labelled so it is unclear what authors are specifically referring to from the text in relation to interpreting the images. Specific cracks/features of interest should be labelled on the image. The last few microscopy samples are poorly explained/integrated with the text and there are no images demonstrating fractures/failures of the part.
Conclusion:
The authors use qualitative words such as “slight impact” whereas quantitative measures would provide better comparisons to justify claims.
Some conclusions, such as conclusion 3 was not directly supported by the study or cited to justify the claim and other possible interpretations of the data.
When referring to printing voids, was there any quantification of the number/size of printing voids in different parts to justify claims of their affects on trends?